



# Worldwide version-controlled database of glacier thickness observations

Ethan Welty[1,2], Michael Zemp[2], Francisco Navarro[3], Matthias Huss[4,5,6], Johannes J. Fürst[7], Isabelle Gärtner-Roer[2], Johannes Landmann[2,4,5], Horst Machguth[6,2], Kathrin Naegeli[8,2], Liss M. Andreassen[9], Daniel Farinotti[4,5], Huilin Li[10], and GlaThiDa Contributors[*]

[1]Institute of Arctic and Alpine Research (INSTAAR), University of Colorado Boulder, United States
[2]World Glacier Monitoring Service (WGMS), University of Zürich, Switzerland
[3]Escuela Técnica Superior de Ingenieros de Telecomunicación, Universidad Politécnica de Madrid, Spain
[4]Laboratory of Hydraulics, Hydrology and Glaciology (VAW), ETH Zürich, Switzerland
[5]Swiss Federal Institute for Forest, Snow and Landscape Research (WSL), Switzerland
[6]Department of Geosciences, University of Fribourg, Switzerland
[7]Department of Geography, University of Erlangen-Nuremberg (FAU), Germany
[8]Institute of Geography, University of Bern, Switzerland
[9]Norwegian Water Resources and Energy Directorate (NVE), Oslo, Norway
[10]Cold and Arid Regions Environmental and Engineering Research Institute, Chinese Academy of Sciences, China
[*]A list of additional contributors and their affiliations appears at the end of the paper.

**Correspondence:** Ethan Welty (ethan.welty@colorado.edu)

**Abstract.**

Although worldwide inventories of glacier area have been coordinated internationally for several decades, a similar effort for glacier ice thicknesses was only initiated in 2013. Here, we present the third version of the Glacier Thickness Database (GlaThiDa v3), which includes 3 854 279 thickness measurements distributed over more than 3 000 glaciers worldwide. Over-
all, 14 % of global glacier area is now within 1 km of a thickness measurement – a significant improvement over GlaThiDa v2, which covered only 6 % of global glacier area, and only 1 100 glaciers. In this paper, we summarize the sources and compilation of glacier thickness data and the spatial and temporal coverage of the resulting database. In addition, we detail our use of open-source metadata formats and software tools to describe the data, validate the data against this description, and track changes to the data following modern data management best-practices. Archived versions GlaThiDa are
available from the World Glacier Monitoring Service (https://doi.org/10.5904/wgms-glathida-2019-03; GlaThiDa Consortium (2019)) and the development version (v3.1.0-rc1, from which this manuscript draft is generated) is available on GitLab (https://gitlab.com/wgms/glathida/-/tree/v3.1.0-rc1).

## 1    Introduction

A central challenge of glaciology is assessing the distribution and total ice volume of the world's glaciers. Increasingly detailed
and globally-complete inventories of the world's glaciers (WGMS and NSIDC, 2012; GLIMS and NSIDC, 2018; RGI Consortium, 2017) have been compiled with great effort over the last few decades. However, these inventories have been limited to





glacier extent and surface elevation. The Glacier Thickness Database (GlaThiDa), launched by the World Glacier Monitoring Service (WGMS, https://wgms.ch) and supported by the International Association of Cryospheric Sciences (IACS) Working Group on Glacier Ice Thickness Estimation (https://cryosphericsciences.org/activities/ice-thickness), complements these existing efforts by compiling and publishing a freely-accessible database of glacier thickness observations (Gärtner-Roer et al., 5  2014).

Knowing the thickness of glacier ice is critical for anticipating the rate and timing of glacier retreat and disappearance, the subsequent effects on local and regional hydrologic cycles and global sea level, and the associated environmental and social impacts. As the only worldwide repository of its kind, GlaThiDa plays an important role in local, regional, and global studies of glaciers, glacier ice volumes, and their potential sea-level rise contributions (e.g. Thorlaksson, 2017; Farinotti et al., 2017, 2019; 10  Meyer et al., 2018; Fischer, 2018; Ayala et al., 2019; Werder et al., 2020). While the Ice Thickness Models Intercomparison eXperiment (ITMIX; Farinotti et al., 2017) only used GlaThiDa v1 (WGMS, 2014) to calibrate one of the participating models, it helped garner support and data for GlaThiDa v2 (WGMS, 2016), which was subsequently used to calibrate ice thickness models and evaluate model performance for an ensemble-based estimate of the thicknesses of all glaciers on Earth (Farinotti et al., 2019).

15  GlaThiDa v3 represents a major step forward for the database. We have more than doubled the spatial coverage and more than quadrupled the number of observations relative to v2, released in 2016, adding over 3 million thickness measurements either submitted by researchers (46 % of new measurements) or imported from the IceBridge Data Portal (54 %, https://nsidc. org/data/icebridge). In addition to summarizing the spatial and temporal coverage of the database, we present a case study on how simple open-source metadata formats and software tools can be used to implement modern data management practices. 20  In the sections that follow, we describe a development environment for data – based on universal text-based file formats and the distributed version-control system `git` (Chacon and Straub, 2014) – that maximizes data access and interoperability, automatically tracks and archives every change made to the dataset, continuously validates the structure and contents of the data, and facilitates dialogue with (and bug reports by) data users.

## 2 Methods and data

### 2.1 Data sources

#### 2.1.1 Data compilation

Since the release of GlaThiDa v1 in 2014 (Gärtner-Roer et al., 2014), which focused on gathering glacier mean and maximum thickness estimates from published literature, a large number of thickness measurements have been submitted by members of the research community in response to two calls for data, one for version 2 in 2016 and another for version 3 in 2018. In all, 30  researchers from institutions in Europe, North and South America, Oceania, and Asia have contributed data from Antarctica, Africa (Kenya, Tanzania), Asia (China, Georgia, India, Kazakhstan, Kyrgyzstan, Mongolia, Nepal, Russia, Tajikistan), Eu-





rope (Austria, Germany, Greenland, Svalbard, Iceland, Italy, Norway, Sweden, Switzerland), Oceania (New Zealand), North America (Canada, United States), and South America (Bolivia, Chile, Colombia, Peru).

Alongside these manual data submissions, airborne glacier thickness profiles collected by National Aeronautics and Space Administration (NASA) Operation IceBridge were retrieved from the corresponding National Snow and Ice Data Center

(NSIDC) data portals. Since these campaigns were primarily focused on the Greenland and Antarctic ice sheets, only measurements within Randolph Glacier Inventory (RGI 6.0) glacier outlines (RGI Consortium, 2017) were included in GlaThiDa. These replaced the IceBridge data available in 2014, intersected with Randolph Glacier Inventory (RGI 3.2) glacier outlines (RGI Consortium, 2013), included in GlaThiDa v1.

### 2.1.2 Measurement methods

The surveys in GlaThiDa span the history of glacier ice thickness measurement, and thus include several different survey methods, summarized in Table 1. The most direct methods involve excavating or drilling through the ice. Although these typically produce a precise measurement, they do so only for a single point and at a great expense of time and money; correspondingly, these account for only 0.35 % of surveys in GlaThiDa (here, a "survey" roughly represents one measurement campaign on one glacier). The drilling surveys added in v3 (compiled by Fürst et al., 2018a, Table S3) were carried out in Svalbard in the

1970s through 1990s to map the thermal structure of glaciers (e.g. Jania et al., 1996) or to extract paleoclimate records (e.g. Kotlyakov et al., 2004). No drilling surveys were added in v2.

More often, ice thickness is measured indirectly using geophysical methods. For example, seismic soundings employ the propagation properties of elastic waves to determine the structure of the subsurface (described in Susstrunk, 1951). Although common in the 1950s through 1980s, they are expensive and time-consuming to collect; correspondingly, these account for

only 0.84 % of surveys in GlaThiDa. No seismic surveys were added in v3; of those already in the database, the majority were carried out in the Austrian Alps up until the 1970s (reviewed in Aric and Brückl, 2001). Only one seismic survey – of Tasman Glacier, New Zealand from 1971 (Anderton, 1975) – was added in v2.

The most common geophysical method is radar (i.e. radio detection and ranging, also known as "radio-echo sounding" or "ground-penetrating radar"), which is based on the transmission, reflection, and subsequent detection of radio waves (reviewed

in Schroeder et al., 2020). Radar measurements can be collected quickly, from the ice surface or from airborne platforms, and account for 98.44 % of surveys in GlaThiDa. Of the radar surveys added in v3, a majority are provided by NASA Operation IceBridge (Koenig et al., 2010), which sponsored a series of airborne radar platforms: the Multichannel Coherent Radar Depth Sounder (MCoRDS; Paden et al., 2011, 2018; Shi et al., 2010), High-Capability Airborne Radar Sounder (HiCARS; Blankenship et al., 2017a, b), Pathfinder Advanced Radar Ice Sounder (PARIS; Raney, 2010), and Warm Ice Sounding Explorer (WISE;

Rignot et al., 2013a, b). Other large additions in v3 include terrestrial and aerial radar surveys in Svalbard (compiled by Fürst et al., 2018a, Table S2) from the early 1980s (e.g. Dowdeswell et al., 1984, 1986) to the present day (e.g. Martín-Español et al., 2013; Navarro et al., 2014; Lindbäck et al., 2018), and helicopter-borne radar surveys in the Swiss Alps (Rutishauser et al., 2016). Large additions in v2 include extensive terrestrial radar surveys of glaciers in the Italian and Austrian Alps (Fischer et al., 2015a, b).

**Table 1.** Number of glacier surveys and point measurements, interquartile range of point thicknesses (if any), and full range of survey years by survey method. In the database, a "survey" roughly represents one measurement campaign on one glacier, and a "point" represents a single ice thickness measurement (as opposed to a spatial mean).

| Method | Surveys | Points | Thickness (m) | Years |
|---|---|---|---|---|
| Radar (airborne) | 4 624 | 3 064 055 | 104–456 | 1968–2017 |
| Radar (terrestrial) | 412 | 700 066 | 87–330 | 1970–2018 |
| Radar (both or unknown) | 25 | 87 481 | 179–323 | 2006–2016 |
| Seismic | 43 | 31 | 218–440 | 1953–1993 |
| Drilling | 18 | 35 | 40–135 | 1935–2007 |
| Electromagnetic | 2 | 2 611 | 47–86 | 2002–2002 |
| Unknown | 17 | 0 | | |

Geophysical techniques less commonly used for measuring glacier ice thickness include geoelectric (e.g. electrical resistivity tomography) and electromagnetic (e.g. magnetotellurics, controlled-source induction) methods which invert variations in electrical resistivity with depth to map the subsurface. The only examples of these in the database (0.04 %), added in v1, are helicopter electromagnetic surveys of two Cascade Range volcanoes (Finn et al., 2012). The methods of the remaining surveys (0.33 %) are unknown because the original source is either not known or cannot be found. All studies included in GlaThiDa are acknowledged in the database; the studies cited above are only provided as examples.

## 2.2 Data package structure

Equally as important as the data itself is the packaging – the physical representation of the data within files, the design of the metadata that describes the data, and the distribution of the data package to prospective users. Without proper packaging, data is much less likely achieve its full potential. The approach described in the forthcoming sections implements (and extends) the FAIR guiding principles: scientific data should be Findable, Accessible, Interoperable and Reusable for both machines and people (Wilkinson et al., 2016). Our implementation, built on simple text files, was designed to meet the following criteria:

– Widely-supported, open, human- and machine-readable file formats to maximize interoperability and ease of use (Cerri and Fuggetta, 2007).

– Compatible with line-based version control systems like `mercurial` and `git` to automatically track and store changes (Blischak et al., 2016), facilitate collaboration between multiple authors (Ram, 2013), and continuously release new versions as the dataset evolves over time (Rauber et al., 2015).

– Described according to existing metadata standards to facilitate data interpretation, validation, and future contributions (Fowler et al., 2017), as well as reuse by software applications like the Global Terrestrial Network for Glaciers (GTN-G) data browser (http://www.glims.org/maps/gtng).



### 2.2.1 `data/*.csv` The data

The data is structured as three relational database tables ordered in increasing level of detail (Figure 1). The first, overview table (`T`) contains information on the location, identity, and area of the surveyed glacier, the survey method used, and details about the authors and sources of the data. Glacier mean and maximum thickness, estimated from point measurements, are also included when available from data providers. The second table (`TT`) contains any mean and maximum thicknesses, estimated from point measurements, for surface elevation bands. The third table (`TTT`) contains point thickness measurements. All tables include a survey identifier (`GlaThiDa_ID`, unique in `T`) that links entries between the tables, as well as a country code (`POLITICAL_UNIT`) and glacier name (`GLACIER_NAME`) which are replicated in `TT` and `TTT` as a convenience to users. Structural changes since GlaThiDa v1, described in the changelog (Section 2.2.4), have been limited to adding fields (e.g. `PROFILE_ID` in v3 to group point measurements by survey profile) and renaming fields (e.g. `DEM_DATE` to `ELEVATION_DATE` in v3 to clarify that provided surface elevations need not be from a digital elevation model).

Following FAIR principles, the three tables are stored as CSV (comma-separated values) files, a universally-supported text format for representing tabular data. To maximize machine-readability, the files do not contain any non-data content other than a single header line with field names. Data documentation is performed by a separate metadata file, described below.

### 2.2.2 `datapackage.json` The metadata

The structure and content of the data package is described in a single JSON (JavaScript Object Notation) file which conforms to the Frictionless Data Tabular Data Package specification (Walsh et al., 2017). This file contains general metadata like the package's name, version, description, and license (CC-BY-4.0: https://creativecommons.org/licenses/by/4.0), a list of contributors, and links to published source datasets. Crucially, the file also contains a detailed description of both the contents and structure of the tabular data files. In practice, CSV files come in an astonishing number of variants; by making the format and character encoding explicit, we help both software and human users avoid unecessary guesswork (Figure 2).

Each table field is described in turn, including its name and description, the data type it represents (e.g. string, number, or integer), and any constraints on the values it can take (e.g. whether a value is required, falls within a numeric range, or matches a search pattern). In the example in Figure 3, the regular expression (Ahmad, 2018) `^\-?[0-9]*(\.[0-9]{1,7})?$` matches a character string representing any positive or negative number with optionally a decimal point and one to seven decimal places.

Finally, relations within and between tables are defined following relational database nomenclature. A unique key is a field (or set of fields) whose values must be unique for each row in the table. Foreign keys link tables together based on the values of one or more fields: each row of the table must match one (and only one) row in the other ("foreign") table. In the example in Figure 4, each row of table `TTT` (point measurements) is uniquely identified by the combination of a survey identifier (`GlaThiDa_ID`), profile identifier (`PROFILE_ID`), and point identifier (`POINT_ID`). Each of these point measurements is linked to the corresponding row in table `T` (survey overviews) by the value of its survey identifier (`GlaThiDa_ID`), along with the replicated fields for country code (`POLITICAL_UNIT`) and glacier name (`GLACIER_NAME`).





**Table `T`. Glacier thickness: Overview**

*5 141 glacier surveys*

```
GlaThiDa_ID
POLITICAL_UNIT
GLACIER_NAME
GLACIER_DB
GLACIER_ID
LAT
LON
SURVEY_DATE
ELEVATION_DATE
AREA
MEAN_SLOPE
MEAN_THICKNESS
MEAN_THICKNESS_UNCERTAINTY
MAXIMUM_THICKNESS
MAX_THICKNESS_UNCERTAINTY
SURVEY_METHOD
SURVEY_METHOD_DETAILS
NUMBER_OF_SURVEY_POINTS
NUMBER_OF_SURVEY_PROFILES
TOTAL_LENGTH_OF_SURVEY_PROFILES
INTERPOLATION_METHOD
INVESTIGATOR
SPONSORING_AGENCY
REFERENCES
DATA_FLAG
REMARKS
```

**Table `TT`. Glacier thickness: By elevation band**

*412 entries from 41 surveys*

```
GlaThiDa_ID
POLITICAL_UNIT
GLACIER_NAME
SURVEY_DATE
LOWER_BOUND
UPPER_BOUND
AREA
MEAN_SLOPE
MEAN_THICKNESS
MEAN_THICKNESS_UNCERTAINTY
MAXIMUM_THICKNESS
MAX_THICKNESS_UNCERTAINTY
DATA_FLAG
REMARKS
```

**Table `TTT`. Glacier thickness: Point measurements**

*3 854 279 entries from 4 681 surveys*

```
GlaThiDa_ID
POLITICAL_UNIT
GLACIER_NAME
SURVEY_DATE
PROFILE_ID
POINT_ID
POINT_LAT
POINT_LON
ELEVATION
THICKNESS
THICKNESS_UNCERTAINTY
DATA_FLAG
REMARKS
```

**Figure 1.** Fields and overall contents of the three database tables `T`, `TT`, and `TTT`. Detailed field descriptions are provided in the metadata file described in Section 2.2.2.





```
{
  "format": "csv",
  "mediatype": "text/csv",
  "encoding": "utf-8",
  "dialect": {
    "header": true,
    "delimiter": ",",
    "lineTerminator": "\n",
    "quoteChar": "\"",
    "doubleQuote": true
  },
  "schema": {
    "missingValues": [""]
  }
}
```

**Figure 2.** Sample JSON from `datapackage.json` specifying the file format (format: "csv"), character encoding (encoding: "utf-8"), and structure of the data files – for example, that the first line of each file contains field names (header: true), values are separated by a comma (delimiter: ","), and missing values are exclusively represented by empty strings (missingValues: [""]).

```
{
  "name": "POINT_LAT",
  "title": "Point latitude (°, WGS 84)",
  "description": "Latitude in decimal degrees (°, WGS 84),
    with up to seven decimal places. Positive values indicate the northern
    hemisphere and negative values indicate the southern hemisphere.",
  "type": "number",
  "constraints": {
    "required": true,
    "minimum": -90,
    "maximum": 90,
    "pattern": "^\\-?[0-9]*(\\.[0-9]{1,7})?$"
  }
}
```

**Figure 3.** Sample JSON from `datapackage.json` specifying the name, title, description, data type, and value constraints for the field `POINT_LAT` in table `TTT`.





```
{
  "uniqueKeys": [
    ["GlaThiDa_ID", "PROFILE_ID", "POINT_ID"]
  ],
  "foreignKeys": [
    {
      "fields": ["GlaThiDa_ID", "POLITICAL_UNIT", "GLACIER_NAME"],
      "reference": {
        "resource": "T",
        "fields": ["GlaThiDa_ID", "POLITICAL_UNIT", "GLACIER_NAME"]
      }
    }
  ]
}
```

**Figure 4.** Sample JSON from `datapackage.json` specifying the unique keys and foreign keys for table `TTT`.

### 2.2.3  `README.md` The storefront

The data package is fully described in `datapackage.json`, but the JSON format may not be familiar or welcoming to some users. Therefore, we dynamically generate a more human-readable version from the contents of `datapackage.json`. The resulting `README.md` is a text file structured with Markdown, a widely-supported markup language (Gruber, 2004). As a

5 result, it is both easy to read as plain text and readily converted to other formats such as HTML (Hypertext Markup Language) or PDF (Portable Document Format). The choice of file formats increases user access while their shared JSON origin eliminates the risks and overhead associated with manually maintaining multiple files.

### 2.2.4  `CHANGELOG.md` The historian

All notable changes made to the data or metadata are recorded in a chronological list formatted in Markdown, `CHANGELOG.md`.

10 This includes the update or removal of existing data records, additions of new data records, and changes to the file structure or data schema. The goal is to provide a variety of user groups with important information about the history of the dataset. Future maintainers can review past changes, developers can evaluate whether and how to update their processing chain based on structural changes, and users can discover what data have been added or updated since the last version.

### 2.3  Product development cycle

15 The Glacier Thickness Database (GlaThiDa) is a community effort that grows as more data is collected. For an evolving dataset like ours, the ability to revise and review collaboratively – to track changes and share those changes with others – is of





great benefit to the communities that contribute to, maintain, and use the data. The development environment should therefore support the following activities:

– Receive, review, and discuss issues with the dataset from and with the community.

– Automatically track all changes made to the dataset by a distributed team of contributors.

– Continuously validate the dataset as changes are made.

– Release new versions on a rolling basis, to be archived – with a unique DOI (digital object identifier) — for distribution, citation, and safekeeping in a scientific data repository (Paskin, 2005).

### 2.3.1 Tooling

To achieve the goals listed above, we have adopted tools, widely used for open-source software development, for open-data
development. In our case, the dataset is stored as a file repository managed by the distributed version-control system `git`, and hosted on GitLab (https://gitlab.com), an open-source equivalent of GitHub (https://github.com), the popular online platform for collaborative software development. The underlying `git` software tracks changes (or "commits"), while GitLab provides interactive tools for garnering and managing input from the community. "Issues", which can be posted by anyone with a free account, track bug reports, feature requests, and other community dialogue. "Releases" tag a snapshot of the dataset, at any
stage in the development cycle, as a numbered version. These snapshots can then be assigned a DOI and placed in a scientific data repository for citing and safekeeping.

Version control systems like `git` are line-based; that is, they track changes to text files on a line-by-line basis. Storing all data and metadata as text files, rather than binary files, allows us to automatically track all changes to the dataset. When fixes are made to existing data records, the change consists of the updated lines. When new records are added, the change consists
of the appended lines. In this way, we avoid making a new copy of a file each time a change is made. Only versions published for download by users are compressed to a binary format, to reduce bandwidth.

### 2.3.2 Versioning

The project follows the Semantic Versioning Specification (Preston-Werner, 2013) for software, adapted for data. Given a version number `major.minor.patch`, the `major` version is incremented for new data, the `minor` version is incremented
for changes to existing data, and the `patch` version is incremented for changes to metadata only.

Note that our versioning scheme does not communicate compatibility with downstream software dependencies, which is the primary purpose of Semantic Versioning. A proposed software-oriented alternative (Pollock and Walsh, 2017) is to increment the `major` version for incompatible changes (e.g. field removed, field constraint made more restrictive), the `minor` version for backwards-compatible changes (e.g. data added, field constraint made less restrictive), and the `patch` version for backwards-
compatible fixes (e.g. fix data errors, update field description). However, we believe our data-oriented versioning is better



```
def test_has_details_if_survey_method_other():
  df = tables['T']
  mask = df['SURVEY_METHOD'] == 'OTH'
  assert df[mask]['SURVEY_METHOD_DETAILS'].notna().all()
```

**Figure 5.** Sample Python code testing whether SURVEY_METHOD_DETAILS is provided whenever SURVEY_METHOD is "OTH" (other).

aligned with our users and contributors, who are primarily concerned with the addition of new data following each call for submissions.

### 2.3.3 Schema validation

A major benefit of describing the data with machine-readable metadata (i.e. datapackage.json) is the ability to automat-
ically validate the data against this description. This includes relations between tables, uniqueness within tables, and whether field values match field types and constraints: for example, whether dates match the expected format ("YYYYMMDD") and latitude and longitude are numbers within the allowed limits ($[-90, 90]$ and $[-180, 180]$). Furthermore, by using a standard format for the metadata, we can automatically validate the metadata itself against this standard.

That said, all metadata standards have their limits; they cannot express all the possible constraints we may wish to im-
pose on our data. In addition to tests of single fields, validation of GlaThiDa includes tests across multiple fields – for ex-
ample, that country codes (POLITICAL_UNIT) and point measurements (TTT.POINT_LAT, TTT.POINT_LON) are spa-
tially consistent with the provided coordinates for the glacier center point (T.LAT, T.LON), or that survey method details (T.SURVEY_METHOD_DETAILS) are provided if the survey method (T.SURVEY_METHOD) is "other". The latter test, im-
plemented in Python, is shown in Figure 5 as an example.

Continuous Integration (CI) is a standard software development practice wherein changes to the code are verified by au-
tomated tests to detect and fix issues as quickly as possible (Fowler, 2006). In our case, we use CI pipelines integrated into GitLab to automatically validate data and metadata whenever a change is made to the repository, catching issues early in the development cycle and, crucially, before the next release.

## 3   Results and discussion

### 3.1   Spatial and temporal coverage

GlaThiDa v3 is the most comprehensive public database of glacier thickness measurements to date. We have added 3 million new thickness measurements relative to GlathiDa v2, released in 2016 (Table 2). The new data, submitted by researchers or imported from the IceBridge data portal (https://nsidc.org/data/icebridge), includes glaciers in Antarctica, Alaska, Canada, China, Greenland, Kazakhstan, Norway, Svalbard, Switzerland, and Tanzania.





**Table 2.** Total number of rows (surveys, elevation bands, or point measurements) and number of surveys represented in each table, by GlaThiDa version ("v", with the release year in parentheses). Since not all glacier surveys have a mean glacier thickness in table `T`, such surveys are counted separately.

| Table | v1 (2014) | v2 (2016) | v3 (2019) |
|---|---|---|---|
| `T`. Glacier thickness: Overview | | | |
| Surveys (all) | 1 493 | 1 601 | 5 141 |
| Surveys (with mean glacier thickness) | 407 | 504 | 500 |
| `TT`. Glacier thickness: By elevation band | | | |
| Surveys represented | 10 | 33 | 41 |
| Elevation bands | 175 | 376 | 412 |
| `TTT`. Glacier thickness: Point measurements | | | |
| Surveys represented | 948 | 1 080 | 4 681 |
| Point measurements | 759 629 | 820 370 | 3 854 279 |

### 3.1.1 Spatial coverage

To evaluate the spatial coverage of GlaThiDa with respect to the world's roughly 217 000 glaciers (RGI Consortium, 2017), we assigned each survey to glaciers in the Randolph Glacier Inventory (RGI 6.0) by intersecting point measurements and nominal glacier centerpoints with RGI glacier outlines. The result is that 3 054 RGI glaciers have thickness measurements in GlaThiDa

(a large increase from 1 133 RGI glaciers in version 2). Out of 5 141 glacier surveys, only 11 (0.2 %) do not fall within an RGI glacier outline. Of these, most are for glaciers not included in RGI 6.0 – specifically, very small glaciers in the European Alps (Blauschnee and Glacier de Tsarmine, Switzerland; Schwarzmilzferner, Austria) and glaciers that may be considered part of the Antarctic Ice Sheet (Lambert Glacier, Starbuck Glacier, and Scharffenbergbotnen). The remaining do not intersect an RGI outline because either the corresponding RGI outline is incorrect (Nördlicher Schneeferner, Germany) or the glacier has

retreated since the survey was conducted (Columbia Glacier, Alaska).

Figure 6 shows the coverage of intersected RGI glaciers on a world map. Table 3 lists the number and total area of intersected RGI glaciers by glacier region (GTN-G, 2017). While the proportion of intersected glaciers in a region is at most 14 % (region 7: Svalbard and Jan Mayen), the proportional area of intersected glaciers is much higher, up to 77 % (again, for Svalbard and Jan Mayen) – a result of larger glaciers being preferentially selected for measurement. The coverage in Svalbard is so high

in v3 thanks to a recent regional compilation of available measurements (Martín-Español et al., 2015; Fürst et al., 2018b). The regions with the next best area coverage are those with substantial contributions from NASA Operation IceBridge: Arctic Canada, Antarctica, and Greenland. Despite these advances, large gaps persist in GlaThiDa, especially throughout Asia, the Russian Arctic, and the Andes. Future efforts should be aimed at increasing spatial coverage and regional representation, both by performing new measurements and by conducting literature surveys and calls for data in underrepresented languages and

regions of the world.

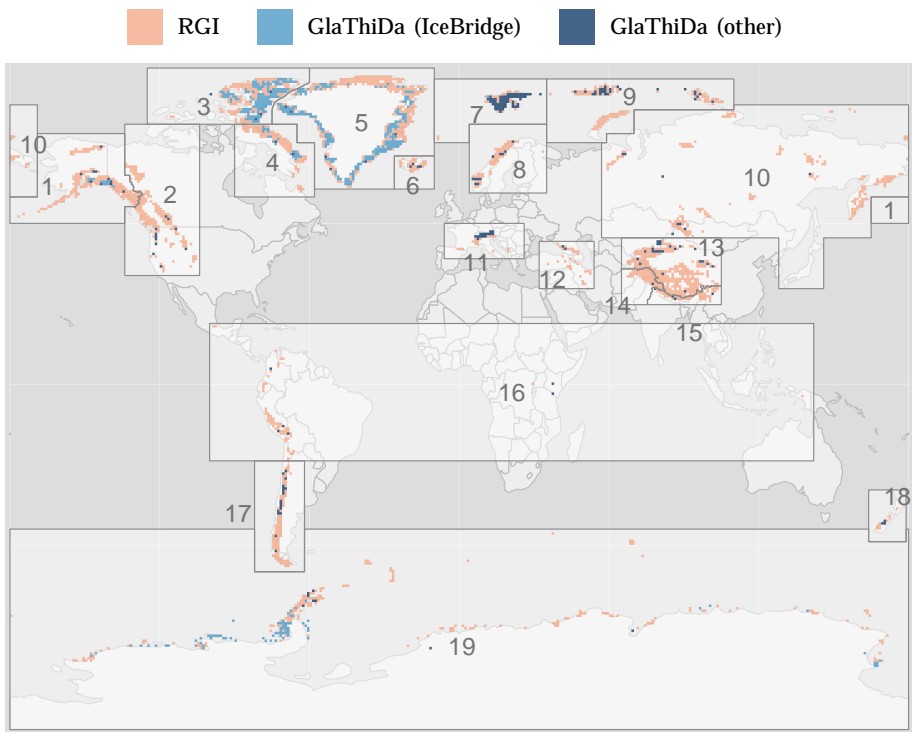

**Figure 6.** Map comparing GlaThiDa coverage to global glacier coverage according to the Randolph Glacier Inventory (RGI 6.0). Each grid cell represents $78.7\,\mathrm{km} \times 78.7\,\mathrm{km}$ (roughly $1\,° \times 1\,°$) in a cylindrical equal-area map projection. Light blue cells contain GlaThiDa data from IceBridge, while the overlaying dark blue pixels contain GlaThiDa data from other sources. Numbered grey polygons correspond to the glacier regions (GTN-G, 2017) listed in Table 3.

Overall, RGI glaciers with at least one thickness measurement account for $40\,\%$ ($299\,141\,\mathrm{km}^2$) of the global RGI area of $746\,092\,\mathrm{km}^2$ (RGI Consortium, 2017). More specifically, $36\,\%$ of the area of these surveyed glaciers ($102\,030\,\mathrm{km}^2$), and $14\,\%$ of the global area, is within $1\,\mathrm{km}$ of a thickness point measurement located on the same glacier. Although this represents a significant improvement over GlaThiDa v2 ($6\,\%$), this nevertheless means that the thickness of the vast majority of global
5   glacier area must still be estimated through extrapolation, scaling methods (reviewed in Bahr et al., 2015), or models (reviewed in Farinotti et al., 2017).

The spatial coverage of point thickness measurements varies greatly by glacier. While $14\,\%$ of glaciers in GlaThiDa with point measurements have more than 100 point measurements per $\mathrm{km}^2$ on average, $50\,\%$ have fewer than 18 points (Figure 7). Although measurements are often sparse with respect to total glacier area, measurements tend to be well distributed across
10   glacier surface elevations, since measurements are often collected along longitudinal profiles. Dividing each glacier into $100\,\mathrm{m}$ elevation bands, $50\,\%$ of glaciers with point measurements have measurements in at least half of their elevation bands and $11\,\%$ have measurements in all of their elevation bands (Figure 8). Glaciers with few measurements but well distributed along





**Table 3.** GlaThiDa coverage for each glacier region mapped in Figure 6. The count and total area (km$^2$) of RGI glacier outlines with at least one (point, elevation band, or glacier-wide) thickness is listed for GlaThiDa v2 (2016) and v3 (2019) alongside the total for all RGI glaciers.

| Region | Name | v2 (count) | v3 (count) | RGI (count) | v2 (km$^2$) | v3 (km$^2$) | RGI (km$^2$) |
|---|---|---|---|---|---|---|---|
| 1 | Alaska | 14 | 41 | 27 108 | 4 469 | 21 141 | 86 725 |
| 2 | Western Canada and USA | 38 | 45 | 18 855 | 118 | 142 | 14 524 |
| 3 | Arctic Canada, North | 239 | 476 | 4 556 | 57 783 | 72 351 | 105 111 |
| 4 | Arctic Canada, South | 24 | 251 | 7 415 | 10 633 | 13 943 | 40 888 |
| 5 | Greenland Periphery | 295 | 1 361 | 20 261 | 51 290 | 63 594 | 130 071 |
| 6 | Iceland | 4 | 4 | 568 | 2 161 | 2 161 | 11 060 |
| 7 | Svalbard and Jan Mayen | 79 | 232 | 1 615 | 9 582 | 26 318 | 33 959 |
| 8 | Scandinavia | 99 | 103 | 3 417 | 800 | 813 | 2 949 |
| 9 | Russian Arctic | 22 | 32 | 1 069 | 3 606 | 5 716 | 51 592 |
| 10 | Asia, North | 20 | 61 | 5 151 | 82 | 144 | 2 410 |
| 11 | Central Europe | 128 | 175 | 3 927 | 498 | 768 | 2 092 |
| 12 | Caucasus and Middle East | 2 | 3 | 1 888 | 5 | 36 | 1 307 |
| 13 | Asia, Central | 48 | 79 | 54 429 | 1 466 | 1 401 | 49 303 |
| 14 | Asia, South West | 0 | 1 | 27 988 | 0 | 17 | 33 568 |
| 15 | Asia, South East | 8 | 8 | 13 119 | 69 | 98 | 14 734 |
| 16 | Low Latitudes | 6 | 9 | 2 939 | 12 | 29 | 2 341 |
| 17 | Southern Andes | 35 | 39 | 15 908 | 733 | 735 | 29 429 |
| 18 | New Zealand | 3 | 3 | 3 537 | 112 | 112 | 1 162 |
| 19 | Antarctic and Subantarctic | 69 | 131 | 2 752 | 76 481 | 89 622 | 132 867 |
| | World | 1 133 | 3 054 | 216 502 | 219 900 | 299 141 | 746 092 |

their length can still be very useful for validating modeled ice thicknesses (Castellani, 2019), which are often computed along longitudinal ice-flow lines (reviewed in Farinotti et al., 2017).

### 3.1.2 Temporal coverage

The glacier thickness surveys in GlaThiDa span the years 1935–2018 (Table 1). This wide range enables comparisons through time for those glaciers with repeat surveys – such as the example in Figure 9. However, it also complicates comparisons between glaciers with different survey dates. Ideally, modeled ice thicknesses are evaluated against known ice thicknesses coincident with the glacier outlines, surface elevations, and other time-varying data used to initialize the model (Farinotti et al., 2017). For large-scale analysis, however, this is rarely possible. For example, the median survey year for RGI glacier outlines is 2002, a decade earlier than the 2012 median for GlaThiDa surveys (Figure 10), and the offset between surveys in GlaThiDa and their corresponding RGI outlines is 11–17 years (interquartile range). As for surface elevations, the majority (88 %) of point

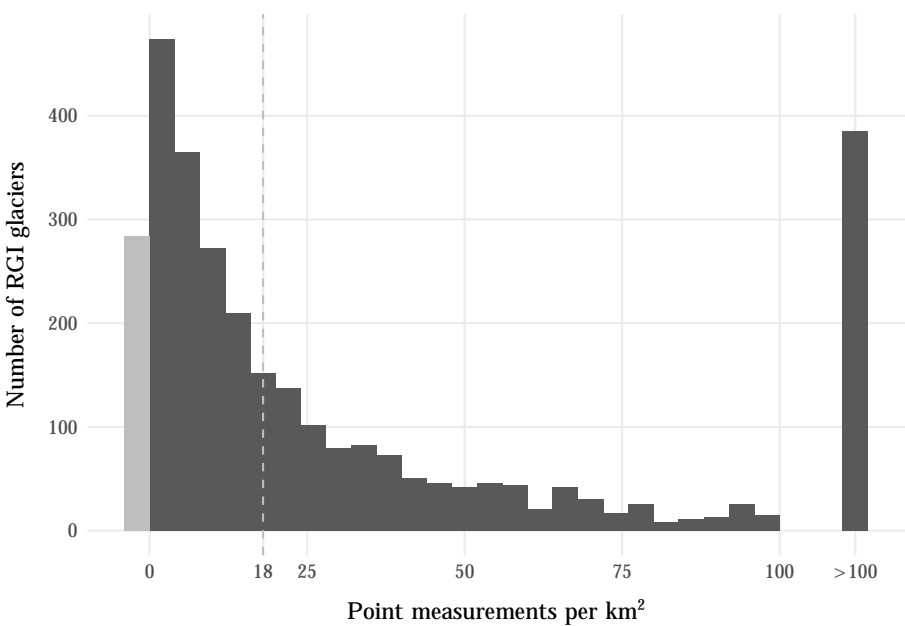

**Figure 7.** Number of RGI glaciers with point measurements in GlaThiDa by the average number of point measurements per km$^2$. The median (18 points per km$^2$) is marked with a grey dashed line. Glaciers with a mean or maximum thickness but no point measurements are shown in the grey bar to the left of 0 per km$^2$.

thickness measurements in GlaThiDa include corresponding surface elevations, 85 % of which were measured the same year as the ice thickness. When available, synchronous surface elevations can be used to calculate the elevation of the glacier bed (independent of the survey date over decades to centuries) and thus the ice thickness relative to a glacier surface surveyed at any other time.

### 3.1.3 Future growth

Our intention is for GlaThiDa to continue to grow and improve as errors are found and fixed, new measurements are made, and more data is found or submitted. Several datasets already published in open data portals (e.g. https://data.npolar.no, https://pangaea.de, https://arcticdata.io, https://nsidc.org, and https://www.usap-dc.org) are slated for inclusion in a future version. However, these account for only a small number of the glacier thickness measurements still missing from GlaThiDa. This is evidenced, for example, by the 460 glacier surveys, pulled from the literature for v1 (Gärtner-Roer et al., 2014), that are still missing the original point measurements from which the reported glacier-wide estimates were derived. Ideally, all ice thickness surveys would be published in open data portals, then added to GlaThiDa for complete coverage in a standard format.

To streamline data aggregation going forward, GlaThiDa may need to be restructured such that data is primarily organized by campaign or dataset, rather than by glacier. The data tables were originally designed to accommodate mean glacier thicknesses pulled from the literature (Gärtner-Roer et al., 2014). Thus, each survey (i.e. each entry in table T, and thus each

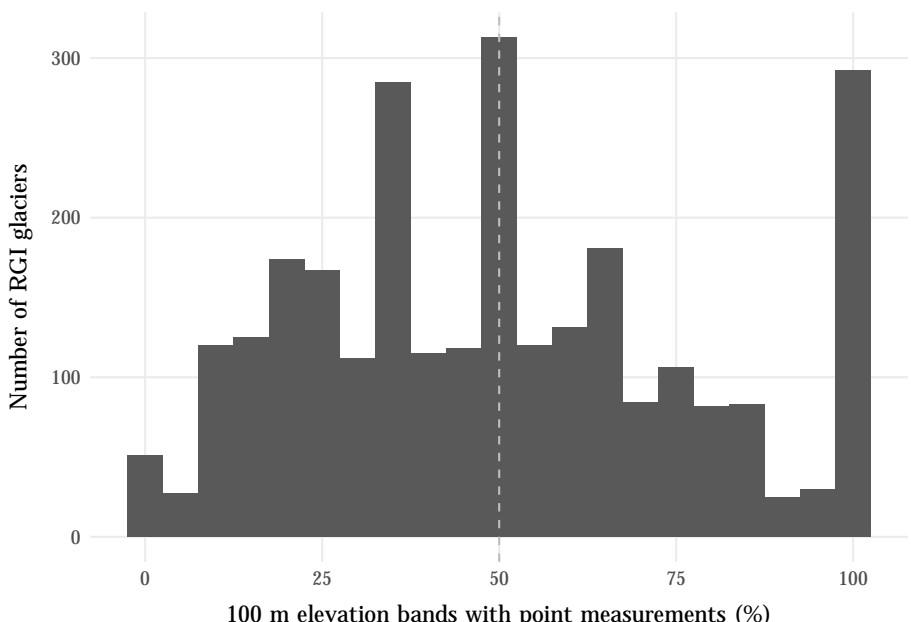

**Figure 8.** Number of RGI glaciers with point measurements in GlaThiDa by the percentage of elevations (discretized into 100 m bands) with one or more point measurements. The median (50 %) is marked with a grey line. Calculated from the surface elevations of point measurements reported in GlaThiDa and the minimum and maximum glacier surface elevations reported in RGI.

`GlaThiDa_ID`) is expected to contain measurements gathered on one visit to one glacier – even though each associated point (i.e. each entry in `TTT`) is also encoded with temporal and spatial coordinates. This data model complicates the addition of large campaigns and introduces confusing redundancy to the database. For example, the six datasets from Operation IceBridge had to be split across 4 124 "glacier surveys" by date and by intersection with RGI glacier outlines.

Operation IceBridge, the source of 61 % of the thickness point measurements in GlaThiDa, are ending operation in 2020. The airborne mission was designed to avoid a gap in measurements between the ICESat satellite (2003–2009) and its successor ICESat-2, which was launched in 2018. However, the ICESat satellites only measure surface elevation, not ice thickness, therefore ending a decade-long ice thickness campaign. In the absence of a successor to Operation IceBridge, future updates to GlaThiDa may not include as many new measurements as the latest version. However, since RGI 6.0 does not include

glaciers on the Antarctic Peninsula mainland (Huber et al., 2017) and in the McMurdo Dry Valleys (Frank Paul, personal communication, 2020), IceBridge data for those glaciers were not included in GlaThiDa v3; these remaining IceBridge data will be added in a future version.

### 3.2 Thickness uncertainties

The uncertainty of a glacier thickness measurement varies widely with the method used, the characteristics of the site, and

the interpretation of the raw data (reviewed in Gärtner-Roer et al., 2014, Section 3.2). For example, sources of error for radar


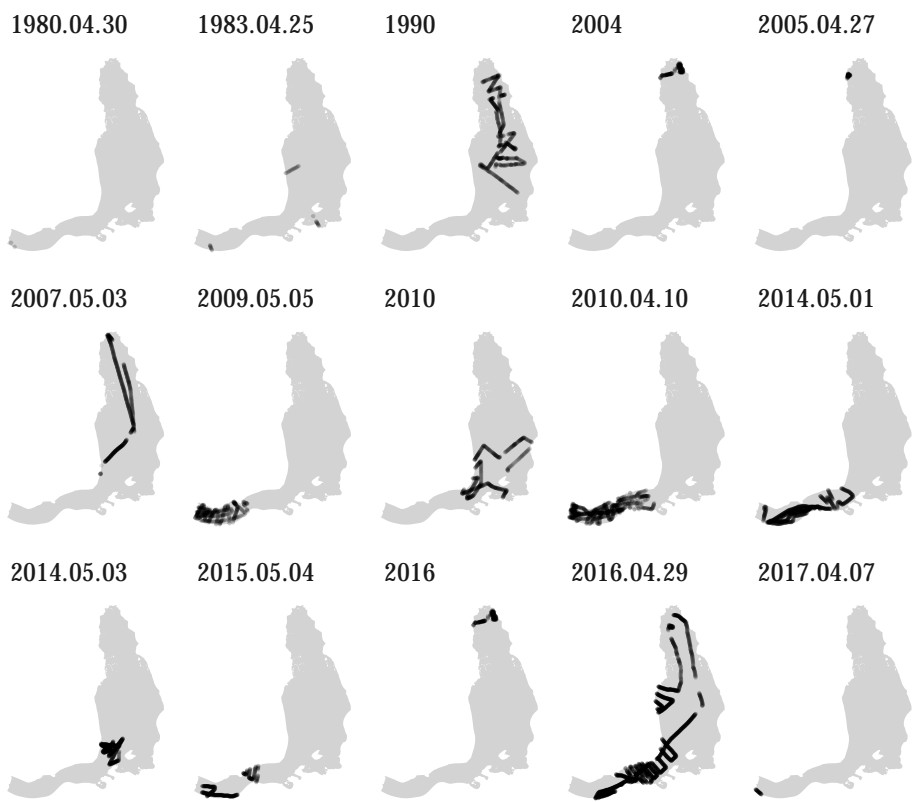

**Figure 9.** All thickness point measurements in GlaThiDa intersecting the RGI outline (RGI60-07.01464) for Kronebreen, Svalbard (78.992°N, 13.384°E), by survey date (or year, if the date is unknown). The corresponding studies referenced in GlaThiDa are Dowdeswell et al. (1984, 1986); Björnsson et al. (1996); Lindbäck et al. (2018); Kristensen et al. (2008); Paden et al. (2018).

measurements (reviewed in Lapazaran et al., 2016a, Section 3.1) include the radio-wave velocity, the timing of the reflection, and migration (inverting for the reflection surface immediately below, rather than to the side), which can fail in or near steep terrain (Welch et al., 1998). Errors in the measurement position (e.g. due to the accuracy, placement, and movement of the GPS receiver) also translate to thickness errors proportional to the local thickness gradient (reviewed in Lapazaran et al., 2016a,

5 Section 3.2). This is a larger issue for older data, especially those transformed from poorly-defined coordinate systems or digitized from printed maps (e.g. Andreassen et al., 2015).

The uncertainty of a spatially-averaged thickness further varies with the adequacy of the interpolation (and extrapolation), the assumed glacier boundary, and most importantly, the spatial coverage of the measurements (reviewed in Lapazaran et al., 2016b). Thickness measurements are typically acquired along sparse profiles, with coverage biased towards gentler terrain.

10 Rarely, if ever, do they approximate a dense grid blanketing the whole glacier. From the law of error propagation, we would expect measurement errors to be smaller for spatial means. In practice, however, the opposite is more commonly the case,

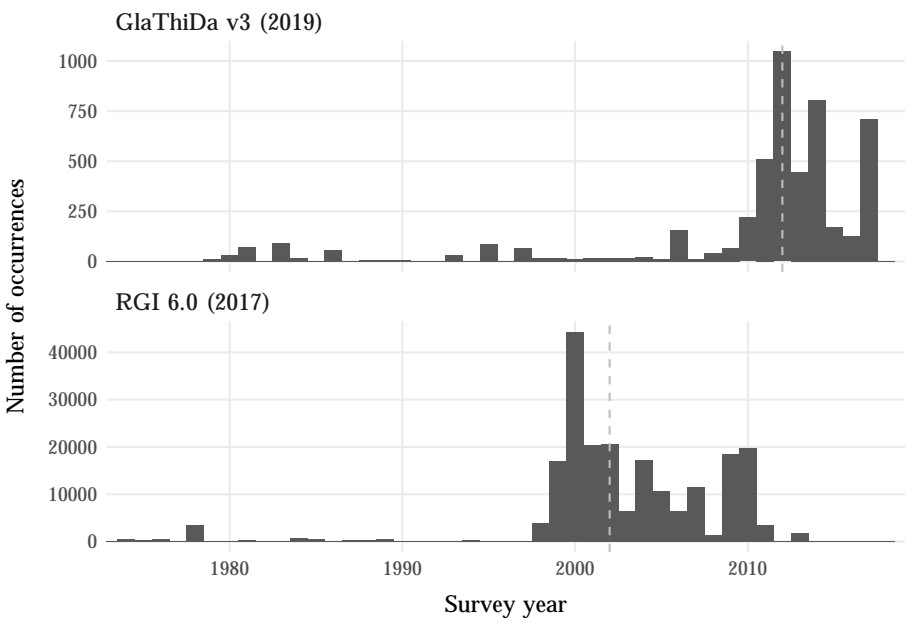

**Figure 10.** Comparison of the distribution of survey years between GlaThiDa (top) and RGI (bottom). Medians are marked with grey dashed lines. Only survey years since 1975 are shown.

since measurement errors are often partly spatially dependent (Martín-Español et al., 2016), rather than truly random, and point coverage is often far from ideal.

   A fraction of glacier thicknesses in GlaThiDa were published with uncertainty estimates: 26 % of mean glacier thicknesses (drawn from about 35 studies, based on the listed references), 19 % of mean elevation-band thicknesses (drawn from 4 studies),

and 40 % of point thicknesses (drawn from 51 studies). Figure 11 compares the distribution of the percent uncertainties ($100\,\%\times$ uncertainty/thickness) reported for each thickness type. The reported uncertainties are significantly lower for point thicknesses than for spatial means – an interquartile range of 3.1–5.5 % of the measured value for points versus 9.9–22.8 % and 20– 50 % for glacier and elevation band thicknesses, respectively. Whether or not the uncertainties reported by these studies are correct, clearly the interpolation errors were deemed to outweigh any benefit gained from averaging out random (spatially-

independent) measurement errors. For thicknesses in GlaThiDa without uncertainties, the distributions in Figure 11 provide a first-order estimate of uncertainty for each thickness type.

   In practice, the statistical definition of the uncertainties reported in GlaThiDa are likely to vary considerably. For example, many studies do not specify whether or not a reported error is the standard deviation. Others may not necessarily provide a full error estimate, but rather the "resolution of the measurements", as in the case of some IceBridge datasets (MCoRDS, HiCARS,

and PARIS). As pointed out by Martín-Español et al. (2016), based in part on two studies included in GlaThiDa (Pettersson et al., 2011; Saintenoy et al., 2013), errors for spatial means ($e$) in the literature can vary by orders of magnitude between two extreme assumptions: (underestimate) local errors ($e_i$) are spatially independent, such that $e = \bar{e}_i/\sqrt{n}$ (where $\bar{e}_i$ is the mean

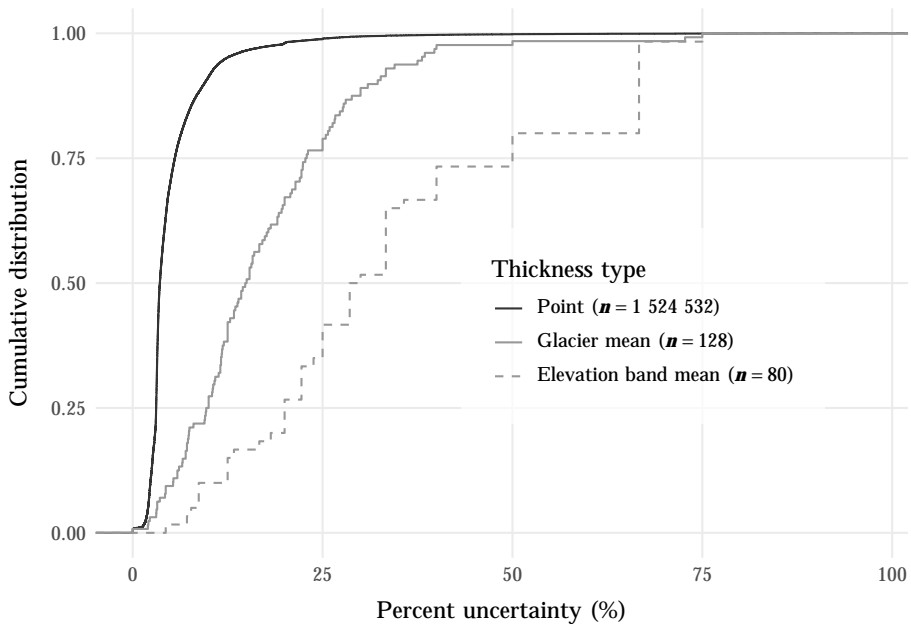

**Figure 11.** Cumulative distribution of percent ice thickness uncertainties ($100\% \times$ uncertainty/thickness) by thickness type: point (solid black line), glacier mean (grey solid line), and elevation band mean (dashed grey line). For example, about three quarters of glacier mean thicknesses report uncertainties of 25 % of the thickness or better.

of the local errors and $n$ is their number), or (overestimate) local errors are linearly dependent, such that $e = \bar{e}_i$ (i.e. the mean of the local errors is taken as the error of the mean). As a consequence, we intend to tighten reporting requirements and flag statistically non-conforming uncertainties in future versions of GlaThiDa.

### 3.3 Data management

5 ### 3.3.1 Error detection

As recorded in the changelog, we fixed a large number of errors introduced in previous versions and in the initial compilation of the current version. Many were trivial to fix once discovered, others required reviewing published literature and datasets, checking original submissions, and if necessary, corresponding with the data provider. Most errors were detected by the suite of automated validation tests described in Section 2.3.3. Checks of field-level constraints identified missing values in required
10 fields, duplicate values in unique fields, out-of-range values in numeric fields, invalid characters in text fields, invalid values in enumerable fields, and future or non-existent dates in date fields. Checks of table-level constraints identified duplicate values in unique keys (e.g. duplicate combinations of survey and point identifiers in `TTT`), and missing values in foreign keys (e.g. survey identifiers in `TT` and `TTT` missing in `T`). More complex tests identified missing values that were required by logical implication (e.g. thickness missing when thickness uncertainty provided), values that were invalid by logical implication (e.g. glacier





identifier not present in the glacier database to which it refers), and values that were invalid by spatial implication (e.g. glacier coordinates outside assigned country or far from associated point measurements). Additional tests will necessarily need to be added proactively as the data evolves, and retroactively as unforeseen errors are introduced and later discovered.

### 3.3.2 Data storage and version control

5 As the number of changes grows over time, `git` repositories inevitably grow in size. Cloud-storage hosts place storage limits on free accounts: GitLab.com limits repositories to 10 gigabytes (GB) compressed; GitHub.com limits repositories to 100 GB and each file to 100 megabytes (MB) uncompressed. To keep repositories small in the presence of large files, `git-lfs` (git Large File Storage) was developed to track files in the repository while storing them externally (Carlson and Schneider, 2019). However, whether the file is binary or text, a new copy of the file is made for each change – no matter how small the change. By 10 storing our data as text files in the repository, changes are stored incrementally, which imparts significant storage benefits. At the time of writing, the repository is only 46.4 MB compressed despite 103 changes, including 8 versions of `TTT.csv` (294 MB uncompressed, 42.3 MB compressed). Reaching the 10 GB storage limit on GitLab.com would require adding (or changing) roughly one billion point measurements. If this limit were ever reached, it could be lifted by migrating to a self-hosted GitLab installation.

15 Nevertheless, line-based version control systems like `git` are not optimized for tracking changes to tabular data. A change to a single cell is recorded as a change to the entire row, and swapping the order of two columns is recorded as a change of every row in the table (Fitzpatrick, 2013). Changes to tabular data can be described more compactly (and legibly) using specialized syntax (e.g. Tabular Diff Specification; Fitzpatrick, 2014), but these impart no storage benefit unless the underlying version control systems were rewritten to use them to store changes internally. Alternatively, many changes can be described as 20 operations rather than as changes to file content, such as a log of Structured Query Language (SQL) commands to a relational database or the change history tracked by OpenRefine (Hirst, 2013). However, these would require specific software and a strict (and non-standard) workflow to make changes to the data.

## 4 Conclusions

The Glacier Thickness Database (GlaThiDa) has been established as the international data repository for glacier ice thickness 25 observations. Version 3 contains standardized data for more than 3 000 glaciers worldwide, collected from in-situ and airborne measurements. Overall, 14 % of global glacier area is now within 1 km of a thickness measurement, although large regional gaps persist, especially in Asia, the Russian Arctic, and the Andes. Thanks to simple metadata formats and a development environment based on open-source software, GlaThiDa fulfills the FAIR principles (Findable, Accessible, Interoperable and Reusable for both machines and people) and surpasses them with automatic version-control, continuous validation, and an 30 interface for community dialogue. Hosted by the World Glacier Monitoring Service (WGMS), GlaThiDa will continue to serve the glaciological community as a trustworthy dataset.





## 5  Data availability

GlaThiDa is maintained as a `git` repository hosted at https://gitlab.com/wgms/glathida. Bug reports, data submissions, and other issues should be posted to the issue tracker at https://gitlab.com/wgms/glathida/-/issues. Published versions of GlaThiDa – those with an assigned DOI (digital object identifier) – are hosted by the World Glacier Monitoring Service (WGMS) in

Zürich, Switzerland (https://doi.org/10.5904/wgms-glathida-2019-03; GlaThiDa Consortium (2019)). The development version (v3.1.0-rc1, from which this manuscript draft is generated) is available on GitLab (https://gitlab.com/wgms/glathida/-/tree/v3.1.0-rc1).

*Team list.*   The additional contributors listed below helped compile earlier versions of GlaThiDa, performed measurements, processed data, and/or submitted data to GlaThiDa. They are listed below in alphabetical order by last name. This list does not include the authors of published

literature and datasets which were added to GlaThiDa by the authors of GlaThiDa. Affiliations were recorded at the time of contribution, and may not be current. Jakob Abermann – Asiaq Greenland Survey, Greenland | Songtao Ai – Wuhan University, China | Brian Anderson – Victoria University of Wellington: Antarctic Research Centre, New Zealand | Serguei M. Arkhipov – Russian Academy of Sciences: Institute of Geography, Russia | Izumi Asaji – Hokkaido University, Japan | Andreas Bauder – Swiss Federal Institute of Technology (ETH)–Zürich: Laboratory of Hydraulics, Hydrology and Glaciology (VAW), Switzerland | Jostein Bakke – University of Bergen: Department of Earth

Sciences, Norway | Toby J. Benham – Scott Polar Research Institute, United Kingdom | Douglas I. Benn – University of Saint Andrews, United Kingdom | Daniel Binder – Central Institute for Meteorology and Geodynamics (ZAMG), Austria | Elisa Bjerre – Technical University of Denmark: Arctic Technology Centre, Denmark | Helgi Björnsson – University of Iceland, Iceland | Norbert Blindow – Institute for Geophysics, University of Münster, Germany | Pascal Bohleber – Austrian Academy of Sciences (ÖAW): Institute for Interdisciplinary Mountain Research (IGF), Austria | Eliane Brändle – University of Fribourg, Switzerland | Gino Casassa – University of Magallanes: GAIA

Antarctic Research Center (CIGA), Chile | Jorge Luis Ceballos – Institute of Hydrology, Meteorology and Environmental Studies (IDEAM), Colombia | Julian A. Dowdeswell – Scott Polar Research Institute, United Kingdom | Felipe Andres Echeverry Acosta | Hallgeir Elvehøy – Norwegian Water Resources and Energy Directorate (NVE), Norway | Rune Engeset – Norwegian Water Resources and Energy Directorate (NVE), Norway | Andrea Fischer – Institute of Interdisciplinary Mountain Research (IGF), Austria | Mauro Fischer – University of Fribourg, Switzerland | Gwenn E. Flowers – Simon Fraser University: Department of Earth Sciences, Canada | Erlend Førre – University of Bergen:

Department of Earth Sciences, Norway | Yoshiyuki Fujii – National Institute of Polar Research, Japan | Mariusz Grabiec – University of Silesia in Katowice, Poland | Jon Ove Hagen – University of Oslo, Norway | Svein-Erik Hamran – University of Oslo, Norway | Lea Hartl – Institute of Interdisciplinary Mountain Research (IGF), Austria | Robert Hawley – Dartmouth College, United States | Kay Helfricht – Institute of Interdisciplinary Mountain Research (IGF), Austria | Elisabeth Isaksson – Norwegian Polar Institute, Norway | Jacek Jania – University of Silesia in Katowice, Poland | Robert W. Jacobel – Saint Olaf College: Physics Department, United States | Michael Kennett

– Norwegian Water Resources and Energy Directorate (NVE), Norway | Bjarne Kjøllmoen – Norwegian Water Resources and Energy Directorate (NVE), Norway | Thomas Knecht – University of Zürich, Switzerland | Jack Kohler – Norwegian Polar Institute, Norway | Vladimir Kotlyakov – Russian Academy of Sciences: Institute of Geography, Russia | Steen Savstrup Kristensen – Technical University of Denmark: Department of Space Research and Space Technology (DTU Space), Denmark | Stanislav Kutuzov – University of Reading, United Kingdom | Javier Lapazaran – Universidad Politécnica de Madrid, Spain | Tron Laumann – Norwegian Water Resources and Energy

Directorate (NVE), Norway | Ivan Lavrentiev – Russian Academy of Sciences: Institute of Geography, Russia | Katrin Lindbäck – Norwegian



Polar Institute, Norway | Peter Lisager – Asiaq Greenland Survey, Greenland | Francisco Machío – Universidad Internacional de La Rioja (UNIR), Spain | Gerhard Markl – Institute of Interdisciplinary Mountain Research (IGF), Austria | Enrico Mattea – University of Fribourg: Department of Geography, Switzerland | Kjetil Melvold – Norwegian Water Resources and Energy Directorate (NVE), Norway | Laurent Mingo – Blue System Integration Ltd., Canada | Christian Mitterer – Institute of Interdisciplinary Mountain Research (IGF), Austria | Andri

Moll – University of Zürich, Switzerland | Ian Owens – University of Canterbury: Department of Geography, New Zealand | Finnur Pálsson – University of Iceland, Iceland | Rickard Pettersson – Uppsala University, Sweden | Rainer Prinz – University of Graz: Department of Geography and Regional Science, Austria | Ya.-M.K. Punning – Estonian Academy of Sciences (USSR Academy of Sciences-Estonia): Institute of Geology, Estonia | Antoine Rabatel – University Grenoble Alpes, France | Ian Raphael – Dartmouth College, United States | David Rippin – University of York, United Kingdom | Andrés Rivera – Center for Scientific Studies (CECs), Chile | José Luis Rodríguez

Lagos – Center for Scientific Studies (CECs), Chile | John Sanders – University of California, Berkeley: Department of Earth and Planetary Science, United States | Albane Saintenoy – University of Paris-Sud, France | Arne Chr. Sætrang – Norwegian Polar Institute, Norway | Marius Schaefer – Austral University of Chile: Institute of Physical and Mathematical Sciences (ICFM), Chile | Stefan Scheiblauer – Environmental Earth Observation Information Technology (ENVEO IT GmbH), Austria | Thomas V. Schuler – University of Oslo, Norway | Heïdi Sevestre – University of Saint Andrews, United Kingdom | Bernd Seiser – Institute of Interdisciplinary Mountain Research (IGF),

Austria | Ingvild Sørdal – University of Oslo: Department of Geosciences, Norway | Jakob Steiner – University of Utrecht: Faculty of Geosciences, Netherlands | Peter Alexander Stentoft – Technical University of Denmark: Arctic Technology Centre (ARTEK), Denmark | Martin Stocker-Waldhuber – Technical University of Denmark: Arctic Technology Centre (ARTEK), Denmark | Bernd Seiser – Institute of Interdisciplinary Mountain Research (IGF), Austria | Shin Sugiyama – Hokkaido University: Institute of Low Temperature Science, Japan | Rein Vaikmäe – Tallinn University of Technology, Estonia | Evgeny Vasilenko – Academy of Sciences of Uzbekistan, Uzbekistan | Nat J.

Wilson – Simon Fraser University: Department of Earth Sciences, Canada | Victor S. Zagorodnov – Russian Academy of Sciences: Institute of Geography, Russia | Rodrigo Zamora – Center for Scientific Studies (CECs), Chile.

*Author contributions.*  MZ conceived the project. EW designed and implemented the data package and development environment, with input from MZ. EW prepared the manuscript, with feedback from MZ, FN, MH, JF, IGR, JL, HM, KN, LMA, DF, and HL. MZ, FN, MH, JF, IGR, JL, HM, and KN helped compile earlier versions of GlaThiDa. LMA, DF, and HL co-chaired the International Association of Cryospheric

Sciences (IACS) Working Group on Glacier Ice Thickness Estimation. Additional contributors are named in the *Team list*.

*Acknowledgements.*  This work is a contribution to the International Association of Cryospheric Sciences (IACS) Working Group on Glacier Ice Thickness Estimation (https://cryosphericsciences.org/activities/ice-thickness). EW was funded by IACS, and by the Federal Office of Meteorology and Climatology (MeteoSwiss) within the framework of the Global Climate Observing System (GCOS) in Switzerland. We are grateful to NASA Operation IceBridge and the National Snow and Ice Data Center for making their data publicly available online. We thank

the countless people who contributed to the planning, execution, and processing of glacier thickness measurements included in GlaThida who are not named in the *Team list*. Institutions that funded the acquisition, processing, and archiving of thickness measurements included in GlaThiDa are acknowledged in the database.



*Competing interests.* The authors declare that they have no conflict of interest.



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
