# Peer review of "Worldwide version-controlled database of glacier thickness observations"

_Earth System Science Data, 2020_

## Referee Comment (RC1) · Aparna Shukla (Referee) · 19 Jun 2020

General comments: This research presents a commendable effort by group in managing a huge (> 3000 glaciers) repository of the world-wide glacier thickness data (nearly 4 million). As already mentioned in the manuscript this dataset has immense importance as far as refining the estimates of global ice reserves is concerned. Authors have also tried to incorporate the field measurements in this version of GlaThiDa v3, which further enhances its importance. Overall, data-wise it is a great contribution, however, there is room for improvement. Specific comments: 1. The paper focuses on describing the data attributes, characteristics and sources. In my opinion including some illustrations of the data (for certain regions or may be one per method) and field photographs of the glaciers investigated (one from each region) may be included to

make the paper more interesting. Since it seems to be the first attempt of the authors to incorporate the field-based thickness measurements in the GlaThiDa v3, adding the same would add more value to the manuscript.

2. The uncertainty part can be improved further in the dataset as well as in the manuscript. Certain method can be employed to standardize the uncertainty associated with the data. Thereafter the data may be sorted in terms of associated uncertainty such that the end-user may know the error involved, and they may choose the data accordingly as per the permissible error-limit of their respective applications.

3. The authors discuss the spatial and temporal coverage of the data in sections 3.1.1 and 3.1.2 respectively. Here they mention that there are certain regions across the globe where the data density is scarce. This mandates that it should also be discussed here that what measures can be taken to improve the data density over these regions and how is it planned to add the temporal data. A discussion on how does these regions of low representation affect the overall quality and import of the database.

Technical comments: In general, the manuscript is well-written except for some places (e.g., line 10 of the Introduction section), the sentences are too long and wordy, making it complicated to understand. This should be checked throughout the manuscript. Besides, at places the fonts are different (e.g., P2, L21; P4, L15). If this is on purpose then it is fine but looks quite abrupt and awkward.
* * *

---

## Short Comment (SC1) · 19 Jun 2020

Thank you very much for your comments. We look forward to taking your thoughts into consideration when revising and improving the manuscript.

Regarding your comment "Certain method can be employed to standardize the uncertainty associated with the data", could you clarify what method or class of methods you have in mind?

In regards to "some illustrations of the data" and "field photographs", Figure 6 of the 2014 GlaThiDa paper (https://doi.org/10.1016/j.gloplacha.2014.09.003) has glacier photos alongside summary thickness data. Is this what you had in mind?

---

## Referee Comment (RC2) · Bruce Raup (Referee) · 23 Jun 2020

This paper provides thorough documentation of the GlaThiDa data set and the workflow behind its production. It is well written and organized, and provides links to a number of other key resources. The paper is a valuable contribution to the literature on glacier thickness measurements. I recommend publication after a few minor revisions, as described below.

- Please add a bit of discussion why glacier area being within 1 km of a thickness measurement is important. I understand that this is a good general measure of coverage, and perhaps that is all that is meant. But it seems to be no guarantee that interpolation will be more accurate. For example, there could be a small glacier within 1 km of a

thickness measure on a different glacier, perhaps of greatly different size.

- Page 5, Lines 28/29: Replace the sentence between [ and ] with: "These unique keys can be stored in other tables, where they are called "foreign keys", to link the tables together."

- Page 5, Line 22: "string, number, or integer". I realize his terminology came from the the Frictionless Data Tabular Data Package specification, but "integer" is a subset of "number", so I think "number (float)" or "decimal number" would be better (more precise).

- The regex for the numbers seems to add friction. Shouldn't "number" refer to the standard ascii representation of floating point values, without reference to a language (regular expressions) that most scientists probably don't use? If someone accidentally deleted a character in the regex and didn't notice, it could mess up code that they're using on the data. Inclusion of the regex for number representation seems needlessly complex, and actually a bit dangerous.

- Also, the Ahmad post you cite describes regular expressions as if there is only one flavor. There are multiple versions of regular expressions, which is why I think including machine code in the JSON file that is particular to one flavor isn't a good idea. Or, this could be fixed by stating which flavor it is. See https://en.wikipedia.org/wiki/Comparison_of_regular-expression_engines But that would add even more needless complexity.

- Page 8, Line 3: The word "dynamically" makes it sound like an interactive application. Maybe "automatically" would be better.

- I recommend less metaphorical subheadings in section 2.2, as annotated in the manuscript, for readability by non-native readers of English.

- See other annotations in the manuscript.

Respectfully submitted, Bruce H. Raup

Please also note the supplement to this comment:
https://essd.copernicus.org/preprints/essd-2020-87/essd-2020-87-RC2-
supplement.pdf

**Supplement:**

[revised manuscript text omitted]

---

## Referee Comment (RC3) · Aparna Shukla (Referee) · 23 Jun 2020

Query1: Regarding your comment "Certain method can be employed to standardize the uncertainty associated with the data", could you clarify what method or class of methods you have in mind?

Clarification: What I meant was that the uncertainty part needs to be given more importance. Ice thickness estimated from a particular method (regardless of analyst or region) should follow same method of uncertainty estimation. It may vary across the ice thickness estimation methods as the parameters introducing error in different methods would differ, however, for each method the parameters to be considered while error estimation should be standardized for uniformity in the database.

[Figure]

Query 2: In regards to "some illustrations of the data" and "field photographs", Figure 6 of the 2014 GlaThiDa paper (https://doi.org/10.1016/j.gloplacha.2014.09.003) has glacier photos alongside summary thickness data. Is this what you had in mind?

Clarification: Yes, something of this sort but rather than just including the field photograph of the glacier, it would be better to have photos of the estimation methods in-process.

---

## Referee Comment (RC4) · Anonymous Referee #3 · 29 Jun 2020

General comments

This article describes data set of global glacier thickness observations, the manuscript describess version 3, so it is not a new dataset, but extensively enlarged with IceBridge and other data (number of datapoints from v2 to v3 increased from 820 370 to 3 854 279). This database contains data that has been collected with tremendous effort, and gathering the data into this database is also a very big task, a truly community effort. Description of database is clear and care is taken to explain reasoning behind the selection of the methods and the structure for the database. Glacier thickness data is very useful for both assessment of total volume of glaciers in the world (sea level rise potential) and for the development and application of models to project the future evolution of the glacier volume. The database is very important for facilitating the use of

these observations. The article is well structured and well written with clear objectives, the goals and design of database are clearly described, adhering to modern requirements for continued development, maintenance and accessibility of data. It is not clear to me what use the TT level of the data will have, I think the point measurements the TTT file contains the data that the user will make use of, rather than of elevation bands that are not clearly or uniformly defined.

Specific comments

The text needs thorough editing in some places, see suggestions below.

In the comments below suggestions are made to delete two unnecessary figures (Figures 8, and 11), the information on these figures can be expressed in the text and would shorten and sharpen the article if authors agree to delete these figures.

The titles of all the subsections in sections 2 and 3 need editing, some are too short and misleading, probably relics from the drafting of the article.

Technical comments

Abstract, line 8, not clear what "this description" is referring to. The sentence is not clear, is the data validated, or the format of it?

Abstract, line 9 something missing before GlaThiDa, insert "of" here?.

Page 2, line 6. I find "anticipating" a strange selection of word here, do you mean assessing, or modelling

Page 3, line 7-8, This sentence is not clear and needs editing. What does "intersecting" here mean, is the location of the data points inside a RGI glacier outline?

Page 4, Table 1, what data is the last line, 17 surveys, 0 points and no thickness measurement, why is this included in the table?

Page 4, line 10, add "to" before "achieve"? something missing in sentence

Page 5, line 21, "n" missing in "unnecessary"

Page 9, line 13-14, something missing in sentence, it is strange

Page 13, line 5, not clear what "it" refer to. The sentence is not clear, why is it important to compare glaciers with different survey dates? Do you mean the same glacier that has been measured several times?

Page 13, line 6-7, not clear how ice thicknesses are coincident with glacier outlines, do you mean within glacier outlines?

Page 13, line 7, suggest to add "measurements" after surface elevation (and delete plural s). what other time-varying data are used? Suggest to specify here

Page 13, line 8, what does "large-scale analysis" mean here? Global? It is not clear to what "this" refers to

Page 13, line 10, sentence is not clear, it reads like surveys correspond to outlines, but isn't the thickness measurements that are within RGI outlines, suggest to edit to clarify

Page 13, line 10, suggest to replace "surface elevations" with "surface elevation measurements" everywhere in text.

Page 14, line 2-4 "synchronous surface elevations" is not clear, suggest "surface elevation measurement from the same time as thickness measurements". The remainder of sentence is also not clear, suggest to edit to something like "with bed elevation measurement any surface elevation measurement at later time will provide a thickness measurement"

Page 14, line 5, suggest to edit section title, it is not clear what "growth" is referring to here, probably the database, but this can be clarified.

Page 14, line 9, suggest to replace "from" with "in"

Page 14, line 9, it is not clear what "This" is referring to, suggest to clarify

Page 15, Figure 8, This figure is not very clear and I am not sure if it useful, suggest to delete this figure, text conveying the information in the main text would safe space in paper. The figure caption needs editing if figure remains in paper.

Page 15, lines 5-12, this section needs editing, it appears that operation IceBridge is (not "are" as in line 5) ending, this has provided large amount of data to this database, the speculative sentence "future updates of GlaThiDa may not include as many new measurements as the latest version" should be deleted. Suggest to edit whole paragraph to emphasize the great gain of thickness measurements due to operation Ice-Bridge, but not possible future development

Page 15, line 13, suggest to edit title, the uncertainty is in thickness measurements

Page 16, line 2, it is not clear what is failing, suggest to edit sentence

Page 16, line 5, "larger issue" suggest to edit and replace with "bigger uncertainty"

Page 16, line 7, sentence not clear "adequacy of the interpolation" does not make sense here, it is about the method of interpolation, right? Suggest to edit and turn sentence around, the spatial coverage is most important for the accuracy of spatially-averaged thickness estimate

Page 16, line 10, "approximate dense grid blanketing" is not clear, suggest to replace or delete sentence

Page 16, line 10-11, sentence is not clear, suggest editing

Page 17, Figure 10 Caption, suggest to replace "survey years since 1975" with "surveys after 1975"

Page 17, line 3, suggest to add "measurement" after thickness (and delete plural "es")

Page 17, line 4, "based on listed references" is not clear, edit the sentence to clarify

Page 17, line 6, "thickness type" probably refers to the three different categories in the

data base, but this is not clear in the text, suggest to clarify

Page 17, line 6, suggest to replace "thicknesses" with "thickness measurements"

Page 17, line 6-8 This sentence needs editing, "glacier and elevation band thicknesses" is not clear, probably refers to database categories?

Page 17, line 8-11, This sentence also needs editing, uncertainties are not "correct" but possible "realistic", "deemed to outweigh any benefit gained from averaging out random errors" also not clear and needs clarification

Page 17, line 10-11, this sentence is not clear, not clear how a distribution on Figure provides estimate of uncertainty. "thickness type" is also not a good term, refer to database categories

Page 18, figure 11 is not very useful and needs better explanation and editing of figure caption, suggest to delete figure and convey information of the figure in text and safe space

Page 18, line 5, suggest to edit section title

Page 19, line 5, "number of changes" what is meant here, corrections in the database or additional input?

Page 19, line 5, suggest to add "will" before "grow"

---

## Author Comment (AC1) · 20 Aug 2020

We thank the reviewers for their thoughtful and detailed reviews. We have revised the manuscript accordingly. Below, we quickly summarize the reviewers' general comments (indicated by >) and our responses to them. In the detailed attachment, all of the general and specific comments are responded to in turn. Reviewer suggestions are tagged as either accepted ([Accept]) or rejected ([Reject]), in which case a reason is always given. Locations in the original manuscript are referenced in the format [Page #, Line #].

> Add field photographs of the measurement methods.

We added a new figure with photos illustrating the four main methods for measuring

glacier thickness: Ground-based and aerial ground-penetrating radar, hot water drilling, and seismic reflection.

> Uncertainties should be standardized across the database.

Since the uncertainties are published or submitted to us by many different data providers, and span the history of glacier thickness measurement, we are not in a position to standardize them further. Nevertheless, we believe documenting the lack and heterogeneity of published uncertainties is an important first step towards a common understanding and reporting of observational uncertainties.

> Impacts of poor regional coverage and plans for improved future coverage?

We are an (unfunded) initiative to collect existing measurements, and thus limited to the data that people publish or submit to us. To add to the discussion of these topics, we added a sentence on the impact of uneven regional coverage to Section 3.1.1 (spatial coverage), and the recommendations of an INTAROS report for improving GlaThiDa data coverage to Section 3.1.3 (future growth).

> Clarify the occasional use of a monotype font.

We added a sentence, following the first use of this font, explaining that it will be used throughout for code samples and software, file, table, and field names.

> Discuss importance of glacier area within 1 km of a thickness measurement.

We added and edited several sentences to clarify that it is meant as a useful measure of global data coverage (the choice of 1 km is arbitrary), and that measurements must be located on a glacier for that glacier's area to be counted (since otherwise, the measurements do nothing to improve interpolations for that glacier).

> Regular expressions come in many flavors and should not be applied to numeric fields.

We revised Section 2.2.2 (metadata) to clarify that all regular expressions conform to a

specific syntax (XML Schema), as now required by the metadata format that we follow, and accordingly updated the regular expressions in the metadata file. We also explain that regular expressions are applied to numeric fields only to enforce decimal place limits on their raw (string) values in the data files.

> Why include table TT?

We added a sentence to Section 2.2.1 (data) explaining that this table exists for rare situations when the original point measurements are not available and only elevation band spatial means are available.

> Remove figures 8 and 11.

We removed these figures and moved information from their captions to the text.

> The section titles in Section 2 and 3 are strange.

We updated the titles of sections 2.2.1 - 2.2.4 and 3.1.3 based on reviewer suggestions.

Please also note the supplement to this comment:
https://essd.copernicus.org/preprints/essd-2020-87/essd-2020-87-AC1-supplement.pdf

**Supplement:**

**Reviewer: Aparna Shukla**

[Figure]

**General comments**

> The paper focuses on describing the data attributes, characteristics and sources. In my opinion including some illustrations of the data (for certain regions or may be one per method) and field photographs of the glaciers investigated (one from each region) may be included to make the paper more interesting. Since it seems to be the first attempt of the authors to incorporate the field-based thickness measurements in the GlaThiDa v3, adding the same would add more value to the manuscript. [...] Rather than just including the field photograph of the glacier, it would be better to have photos of the estimation methods in-process.

We have added a new figure with photos illustrating the four main methods for measuring glacier thickness: Ground-based ground-penetrating radar (Johnsons Glacier, Antarctica), aerial ground-penetrating radar (Hansbreen Svalbard), hot water drilling (Rhonegletscher, Switzerland), and seismic reflection (Grubengletscher, Switzerland). We have not added photos of glaciers from each world region, as glaciers from different regions can be indistinguishable.

> The uncertainty part can be improved further in the dataset as well as in the manuscript. Certain method can be employed to standardize the uncertainty associated with the data. Thereafter the data may be sorted in terms of associated uncertainty such that the end-user may know the error involved, and they may choose the data accordingly as per the permissible error-limit of their respective applications. [...] The uncertainty part needs to be given more importance. Ice thickness estimated from a particular method (regardless of analyst or region) should follow same method of uncertainty estimation. It may vary across the ice thickness estimation methods as the parameters introducing error in different methods would differ, however, for each method the parameters to be considered while error estimation should be standardized for uniformity in the database.

Uncertainties are indeed important, but we are not in a position to standardize them further. The uncertainties that we describe in Section 3.2 are those published or submitted to us by many different data providers, and span the history of glacier thickness measurement. We have little control over the methods that were used to estimate these uncertainties. However, we hope to command more leverage on data submissions in the future. Thus we write: [Page 18, Line 2] "As a consequence, we intend to tighten reporting requirements and flag statistically nonconforming uncertainties in future versions of GlaThiDa." Documenting the lack and heterogeneity of published uncertainties is an important first step towards a common understanding and reporting of observational uncertainties.

The data can be sorted and filtered by their uncertainties by using the uncertainties columns in the database tables.

> The authors discuss the spatial and temporal coverage of the data in sections 3.1.1 and 3.1.2 respectively. Here they mention that there are certain regions across the globe where the data density is scarce. This mandates that it should also be discussed here that what measures can be taken to improve the data density over these regions and how is it planned to add the temporal data. A discussion on how does these regions of low representation affect the overall quality and import of the database.

We are not a consortium of data collectors, but rather an (unfunded) initiative to collect existing measurements. Ultimately, we are limited to the data that people submit to us or publish in public repositories. As a result, we had limited ourselves to the following recommendations:

- [Page 11, Line 18] "Future efforts should be aimed at increasing spatial coverage and regional representation, both by performing new measurements and by conducting literature surveys and calls for data in underrepresented languages and regions of the world."

- [Page 14, Line 11] "Ideally, all ice thickness surveys would be published in open data portals, then added to GlaThiDa for complete coverage in a standard format."

However, in light of your feedback, we have added the following:

- [Page 11, Line 18] Added "Poor coverage in these regions necessarily limits the quality of local and global glacier volume assessments and predictions of future change".

- [Page 14, Line 11] Added "An assessment of the Arctic data in GlaThiDa v2 by the Integrated Arctic Observation System (INTAROS, 2018) identified missing observations and concluded that pressure must continue to be placed on research groups to submit their data to the wider community". (see https://intaros.nersc.no/content/report-present-observing-capacities-and-gaps-land-and-cryosphere)

> [e.g. Page 2, Line 21; Page 4, Line 15] At places the fonts are different. If this is on purpose then it is fine but looks quite abrupt and awkward.

To clarify our use of a monospace font, we have added the following:

[Page 2, Line 23] Added "A monospace font is used throughout the manuscript for software packages (e.g. `git`), files (e.g. `datapackage.json`), database tables (e.g. `T`), database table fields (e.g. `POINT_LAT`), and code samples (e.g. Figure 2)."

**Specific comments**

> [e.g. Page 2, Line 10] In general, the manuscript is well-written except for some places the sentences are too long and wordy, making it complicated to understand. This should be checked throughout the manuscript.

[Page 2, Line 10] Changed "While the Ice Thickness Models Intercomparison eXperiment (ITMIX; Farinotti et al., 2017) only used GlaThiDa v1 (WGMS, 2014) to calibrate one of the participating models, it helped garner support and data for GlaThiDa v2 (WGMS, 2016), which was subsequently used to calibrate ice thickness models and evaluate model performance for an ensemble-based estimate of the thicknesses of all glaciers on Earth (Farinotti et al., 2019)." to "The Ice Thickness Models Intercomparison eXperiment (ITMIX; Farinotti et al., 2017) only used GlaThiDa v1 (WGMS, 2014) to calibrate one of the participating models, but it helped garner support and data for GlaThiDa v2 (WGMS, 2016). GlaThiDa v2 was subsequently used to calibrate all participating models and evaluate model performance for an ensemble-based estimate of the thicknesses of all glaciers on Earth (Farinotti et al., 2019).".

Several of the more confusing sentences were pointed out by other reviewers and subsequently rewritten.

**Reviewer: Bruce Raup**

**General comments**

> Please add a bit of discussion why glacier area being within 1 km of a thickness measurement is important. I understand that this is a good general measure of coverage, and perhaps that is all that is meant. But it seems to be no guarantee that interpolation will be more accurate. For example, there could be a small glacier within 1 km of a thickness measure on a different glacier, perhaps of greatly different size.

The metric is simply a means to measure and track global data coverage. It is defined as the total glacier area within 1 km of a thickness measurement "located on the same glacier" (see Section 3.1); glacier area is not included if the measurement is on a neighboring glacier. To clarify these points, we have made the following changes:

- [Page 1, Line 5; Page 19, Line 26] Changed "14% of global glacier area is now within 1 km of a thickness measurement" to "14% of global glacier area is now within 1 km of a thickness measurement (located on the same glacier)".

- [Page 12, Line 2] Added "A better measure of data coverage is the area of a glacier that is within a certain distance of a thickness point measurement located on the same glacier."

- [Page 12, Line 2] Changed "More specifically, 36% of the area of these surveyed glaciers (102 030 km2), and 14% of the global area, is within 1 km of a thickness point measurement located on the same glacier" to "Globally, 36% of the area of all surveyed glaciers (102 030 km2), and 14% of global glacier area, is within 1 km of a measurement".

> The regex for the numbers seems to add friction. Shouldn't "number" refer to the standard ascii representation of floating point values, without reference to a language (regular expressions) that most scientists probably don't use? If someone accidentally deleted a character in the regex and didn't notice, it could mess up code that they're using on the data. Inclusion of the regex for number representation seems needlessly complex, and actually a bit dangerous. Also, the Ahmad post you cite describes regular expressions as if there is only one flavor. There are multiple versions of regular expressions, which is why I think including machine code in the JSON file that is particular to one flavor isn't a good idea. Or, this could be fixed by stating which flavor it is. See https://en.wikipedia.org/wiki/Comparison_of_regular-expression_engines. But that would add even more needless complexity.

Our use of regular expressions on `number` fields is merely a way to enable automated checks of decimal place limits in the data files. Since Frictionless Data's Table Schema specification does not support this constraint natively (see https://github.com/frictionlessdata/specs/issues/641), we opted for a generic and flexible solution: extending the existing `pattern` constraint to non-string fields by checking the pattern against the raw values (i.e. strings) stored in the data files. Independent checks ensure that each field value is a valid "number", "integer", etc (as defined by the Table Schema specification: https://specs.frictionlessdata.io/table-schema/#types-and-formats).

The data files (`data/*.csv`) can be worked with directly, whether or not the JSON metadata (`datapackage.json`) is valid or present. Users can read the JSON metadata (or the friendlier Markdown equivalent), but they have no need to modify it. Mainly, it is a powerful tool for data maintainers, since it powers detailed and automatic data validation, ensuring the data files delivered to users are sound. Regular expressions are a very precise, compact, and machine-readable way of describing constraints on field values. Thanks to automatic tests, any breaking change made by maintainers to any of the files in the data repository (including changes to the regular expressions) would immediately be detected.

Finally, Frictionless Data's Table Schema specification (https://specs.frictionlessdata.io/table-schema/#constraints) has addressed the question of differing regular expression syntax by requiring `pattern` constraints conform to the standard and barebones XML Schema syntax (http://www.w3.org/TR/xmlschema-2/#regexs).

In light of these considerations, we have made the following changes to the text and the data package:

- [Page 5; Line 24] Changed "In the example in Figure 3, the regular expression (Ahmad, 2018) `^\-?[0-9]*(\.[0-9]{1,7})?$` matches a character string representing any positive or negative number with optionally a decimal point and one to seven decimal places" to "In the example in Figure 3, the description informs users that the field values are stored in the data files with 'up to seven decimal places', while the pattern `\-?[0-9]*(\.[0-9]{1,7})?` (a regular expression conforming, as required by the Frictionless Data specification, to the XML Schema syntax; Biron, 2004) makes possible an automated test that this is indeed the case.". Biron, 2004 is a reference to the XML Schema specification (http://www.w3.org/TR/xmlschema-2/#regexs).

- [Figure 2] Changed pattern `^\-?[0-9]*(\.[0-9]{1,7})?$` to `\-?[0-9]*(\.[0-9]{1,7})?`.

- [ `datapackage.json` ; `README.md` ] Updated the regular expressions to conform to XML Schema syntax (and updated our test suite accordingly):

    - Removed unsupported start ( `^` ) and end ( `$` ) anchors (XML Schema requires strings to match the entire regular expression).

    - Replaced unsupported non-capturing ( `(?: )` ) with standard ( `( )` ) groups (these are equivalent for XML Schema, which is only concerned with whether or not strings match the regular expression).

- [ `datapackage.json` ; `README.md` ] Updated field descriptions to include, in words, any `pattern` constraint:

    - Added 'Cannot contain double quotes (") or whitespace other than space' for `[^"\s]+( [^"\s]+)*` .

    - Added 'Cannot contain double quotes (") or whitespace' for `[^"\s]+` .

**Specific comments**

> [Page 1, Line 4] "more than 3000 glaciers": Give precise number.

[Accept] Changed "more than 3000 glaciers" to "roughly 3000 glaciers" – dropping the "more than" marketing term.

There is not a precise number. A "glacier" is an ambiguous unit of measure (e.g. does a retreating branching glacier, or a large icecap with multiple drainage basins, count as multiple glaciers?). Even if we choose RGI glacier boundaries (which are themselves sometimes arbitrary or inconsistent with respect to drainage basins), some of the measurements fall outside the boundaries (i.e. the list in Section 3.1.1).

> [Page 1, Line 5] "within 1 km of a thickness measurement": Why is this important? Does that make interpolation easier?

[Accept] We agree that clarification is needed. Added the following sentence: "Improvements in measurement coverage increase the robustness of numerical interpolations and model extrapolations, resulting in better estimates of regional to global glacier volumes and their potential contributions to sea-level rise.".

> [Page 1, Line 9] insert "of" into "versions GlaThiDa"

[Accept] Inserted "of".

> [Page 2, Line 11] change "only used GlaThiDa v1 to calibrate one of the participating models" to "used GlaThiDa v1 to calibrate only one of the participating models" (?)

[Reject] We believe our original wording more clearly emphasizes the marginal use of GlaThiDa v1 by ITMIX (in contrast to the central use of GlaThiDa v2 by Farinotti et al. 2019).

> [Page 2, Line 16] change "over 3 million" to "approximately 3 million" (over = marketing term)

[Accept] Since "million" already indicates an approximation, changed "over 3 million" to "3 million".

> [Page 3, Line 3] "manual": Manual in what sense?

[Accept] Changed "manual data submissions" to "data submissions". "submission" already suggests that the data was submitted to us, as opposed to harvested by us.

> [Page 4, Line 8] remove "as" in "Equally as important"

[Accept] For additional clarity, changed "Equally as important as the data itself is the packaging – the physical representation [...]" to "Packaging of data is as important as the data themselves. This includes the physical representation [...]".

> [Page 4, Line 9] change "data [is much less likely achieve its full potential]" to "a dataset [...]"

[Accept] For continuity with previous reworked sentence, changed "data is much less likely to achieve its full potential" to "data are much less likely to achieve their full potential".

> [Page 4, Line 10] change "forthcoming" to "following"

[Accept] Changed "forthcoming" to "following".

> [Page 5, Line 1] change header "data/.csv The data" to "Data (data/.csv)"

[Accept] Changed "data/.csv The data" to "Data (data/.csv)".

> [Page 5, Line 2] change "The data is" to "The data are"

[Accept] Changed "The data is" to "The data are".

> [Page 5, Line 15] change header "datapackage.json The metadata" to "Metadata (datapackage.json)"

[Accept] Changed "datapackage.json The metadata" to "Metadata (datapackage.json)".

> [Page 5, Line 18] move "CC-BY-4.0: https://creativecommons.org/licenses/by/4.0" to Table 2

[Accept/Reject] We have removed this parenthetical mention of the license and added it to the "Data availability" section [Page 20, Line 7]:

"GlaThiDa is licensed under Creative Commons Attribution 4.0 International (CC-BY-4.0: \url{https://creativecommons.org/licenses/by/4.0})."

> [Page 5, Line 19] change "Crucially, the file" to "The file"

[Accept] Changed "Crucially, the file" to "The file".

> [Page 5, Line 20] change "an astonishing number of variants" to "a large number of variants"

[Accept] Changed "an astonishing number of variants" to "a large number of variants".

> [Page 5, Line 22] "string, number, or integer". I realize his terminology came from the the Frictionless Data Tabular Data Package specification, but "integer" is a subset of "number", so I think "number (float)" or "decimal number" would be better (more precise).

[Accept] Changed "string, number, or integer" to "string, floating point number, or integer" to be both more precise and consistent with the terminology used in the Frictionless Data Tabular Data Package specification (and thus in `datapackage.json`).

> [Page 5, Line 28-29] change "Foreign keys link tables together based on the values of one or more fields: each row of the table must match one (and only one) row in the other ("foreign") table" to "These unique keys can be stored in other tables, where they are called "foreign keys", to link the tables together."

[Accept] Changed "Foreign keys link tables together based on the values of one or more fields: each row of the table must match one (and only one) row in the other ("foreign") table" to "Unique keys can be stored in other tables (where they are called "foreign keys") to link the tables together.".

> [Page 8, Line 1] change "README.md The storefront" to "Documentation starting point (README.md)"

[Accept] Changed "README.md The storefront" to "Documentation (README.md)".

> [Page 8, Line 3] "dynamically [generate a more human-readable version]": The word "dynamically " makes it sound like an interactive application. Maybe "automatically" would be better.

[Accept] Changed "dynamically" to "automatically".

> [Page 8, Line 8] change "CHANGELOG.md The historian" to "Database history (CHANGELOG.md)"

[Accept] Since not only changes to the "database" are recorded, changed "CHANGELOG.md The historian" to "History (CHANGELOG.md)".

> [Page 12, Line 8] change "per km2" to "/ km2" or spell out km2

[Accept] Changed all instances of "per km2" to "km-2".

> [Page 15, Line 5] change "Operation IceBridge […] are" to "Operation IceBridge […] is"

[Accept] Changed "Operation IceBridge […] are" to "Operation IceBridge […] is".

> [Page 15, Line 10] change "on […] and […]" to "on […] or […]"

[Accept] Changed "on […] and […]" to "on […] or […]".

> [Page 19, Line 25] "more than 3000": Actual number is better than this marketing expression

[Accept] Changed "more than 3000 glaciers" to "3000 glaciers" – dropping the "more than" marketing term and letting the rounding communicate that the number is (and is limited to being) an approximation.

**Reviewer: Anonymous**

**General comments**

> It is not clear to me what use the TT level of the data will have, I think the point measurements the TTT file contains the data that the user will make use of, rather than of elevation bands that are not clearly or uniformly defined.

The `TT` table mainly exists for the rare situations (5 so far, from the literature) when the original point measurements are not available but elevation band summaries are available. To clarify this point, we have made the following change:

[Page 5, Line 6] Added "Although rare, some ice thickness surveys are only available as surface elevation band estimates, their point measurements having been lost or never published.".

> In the comments below suggestions are made to delete two unnecessary figures (Figures 8, and 11), the information on these figures can be expressed in the text and would shorten and sharpen the article if authors agree to delete these figures.

We have removed Figure 8 [Page 15] and Figure 11 [Page 18] and made the following changes to the text:

- [Page 12, Line 11] Inserted "(calculated from the surface elevations of point measurements in GlaThiDa and the minimum and maximum glacier surface elevations in RGI)" after "Dividing each glacier into 100 m elevation bands".

- [Page 17, Line 3-11] Rewrote entire paragraph: "A fraction of glacier thicknesses in GlaThiDa were published with uncertainty estimates: 26% of mean glacier thicknesses in table `T` (drawn from about 35 studies, based on the listed references), 19% of mean elevation-band thicknesses in table `TT` (drawn from 4 studies), and 40% of point thicknesses in `TTT` (drawn from 51 studies). By computing percent uncertainties (100 % × uncertainty / thickness), we can compare the distribution of uncertainties by thickness "types". We find that the reported uncertainties are significantly lower for point thicknesses than for the spatial means – an interquartile range of 3.1–5.5 % of the measured value for points versus 9.9–22.8 % and 20–50 % for glacier and elevation band means, respectively. The uncertainties reported by these studies may or may not be realistic. Nevertheless, the relatively high uncertainties reported for spatial means clearly indicates that, for these studies, interpolation errors outweighed any benefit gained from averaging the spatially-independent errors in the point measurements.

> The titles of all the subsections in sections 2 and 3 need editing, some are too short and misleading, probably relics from the drafting of the article.

- Section 2.2.1: Changed "data/.*csv The data*" to "Data (data/.csv)"

- Section 2.2.2: Changed "datapackage.json The metadata" to "Metadata (datapackage.json)"

- Section 2.2.3: Changed "README.md The storefront" to "Documentation (README.md)"

- Section 2.2.4: Changed "CHANGELOG.md The historian" to "History (CHANGELOG.md)"

- Section 3.1.3: Changed "Future growth" to "Future additions"

**Specific comments**

> [Page 1, Line 8] Not clear what "this description" is referring to. The sentence is not clear, is the data validated, or the format of it?

[Accept] Changed "validate the data against this description" to "validate the data format and content against this metadata description".

"this description" refers to the use of open-source metadata formats and software tools to describe the data. Both the data format and content is validated against this description. Hopefully the changes clarify these points.

> [Page 1, Line 9] Something missing before GlaThiDa, insert "of" here?

[Accept] Inserted "of".

> [Page 2, Line 6] I find "anticipating" a strange selection of word here, do you mean assessing, or modelling.

[Accept] Changed "anticipating" to "predicting".

> [Page 3, Line 7-8] This sentence is not clear and needs editing. What does "intersecting" here mean, is the location of the data points inside a RGI glacier outline?

[Accept] Changed "These replaced the IceBridge data available in 2014, intersected with Randolph Glacier Inventory (RGI 3.2) glacier outlines \citep{rgiconsortium_2013}, included in GlaThiDa v1." to "These replaced the IceBridge data, located within Randolph Glacier Inventory (RGI 3.2) glacier outlines \citep{rgiconsortium_2013}, added to GlaThiDa v1 in 2014.".

> [Page 4, Table 1] What data is the last line, 17 surveys, 0 points and no thickness measurement, why is this included in the table?

[Accept] Removed last line of Table 1.

These are poorly-documented surveys, pulled from the literature, for which we know neither the survey method, survey date, nor have any point measurements (only mean, and sometimes maximum, glacier thickness).

> [Page 4, Line 10] Add "to" before "achieve"? something missing in sentence

[Accept] Changed "is much less likely achieve" to "is much less likely to achieve".

> [Page 5, Line 21] "n" missing in "unnecessary"

[Accept] Changed "unecessary" to "unnecessary".

> [Page 9, Line 13-14] something missing in sentence, it is strange

[Accept] Changed ""Issues", which can be posted by anyone with a free account, track bug reports, feature requests, and other community dialogue" to ""Issues" – which can be posted by anyone with a free account – track bug reports, feature requests, and other community dialogue".

> [Page 13, line 5] not clear what "it" refer to. The sentence is not clear, why is it important to compare glaciers with different survey dates? Do you mean the same glacier that has been measured several times?

[Accept] Changed "This wide range enables comparisons through time for those glaciers with repeat surveys […]. However, it also complicates comparisons between glaciers with different survey dates" to "This wide range of survey dates enables glaciers with repeat surveys ([…]) to be compared over time. However, it also complicates regional and global studies, which must account for thickness measurements spanning multiple years".

> [Page 13, Line 6-7] not clear how ice thicknesses are coincident with glacier outlines, do you mean within glacier outlines?

[Accept] Changed "known ice thicknesses coincident with the glacier outlines" to "measured ice thicknesses coincident in time with any glacier outlines".

> [Page 13, Line 7] suggest to add "measurements" after surface elevation (and delete plural s). what other time-varying data are used? Suggest to specify here

[Accept] Changed "the glacier outlines, surface elevations, and other time-varying data used to initialize the model" to "measured glacier outlines, surface elevations, and other time-varying parameters (e.g. surface mass balance, rates of ice thickness change, and surface velocities) used to initialize the model".

Typically, both surface elevations and glacier outlines (i.e. the line at which glacier thickness reaches 0) are measurements.

> [Page 13, Line 8] what does "large-scale analysis" mean here? Global? It is not clear to what "this" refers to

[Accept] Changed "For large-scale analysis, however, this is rarely possible" to "For analysis spanning many glaciers (or all of the world's glaciers), it is not possible to ensure that all these data are coincident in time".

> [Page 13, Line 10] sentence is not clear, it reads like surveys correspond to outlines, but isn't the thickness measurements that are within RGI outlines, suggest to edit to clarify

[Accept] Changed "their corresponding RGI outlines" to "their spatially coincident RGI outlines".

> [Page 13, Line 10] suggest to replace "surface elevations" with "surface elevation measurements" everywhere in text.

[Reject] It is unclear to us why this would be necessary, since often we are referring to the physical attribute rather than a measurement of it. However, as reported above and below, we have added "measured" or "measurement" to some instances of "surface elevation".

> [Page 14, Line 2-4] "synchronous surface elevations" is not clear, suggest "surface elevation measurement from the same time as thickness measurements". The remainder of sentence is also not clear, suggest to edit to something like "with bed elevation measurement any surface elevation measurement at later time will provide a thickness measurement"

[Accept] Changed "synchronous surface elevations" to "temporally coincident ice thickness and surface elevation measurements".

> [Page 14, Line 5] suggest to edit section title, it is not clear what "growth" is referring to here, probably the database, but this can be clarified.

[Accept] Changed "Future growth" to "Future additions".

> [Page 14, Line 9] suggest to replace "from" with "in"

[Accept] Changed "missing from GlaThiDa" to "missing in GlaThiDa".

> [Page 14, Line 9] it is not clear what "This" is referring to, suggest to clarify data base, but this is not clear in the text, suggest to clarify

[Accept] Changed "This is evidenced, for example, by the 460 glacier surveys, pulled from the literature for v1 (Gärtner-Roer et al., 2014), that are still missing […]" to "For example, 460 glacier surveys pulled from the literature for v1 (Gärtner-Roer et al., 2014) are still missing […]".

> [Page 17, Line 6] suggest to replace "thicknesses" with "thickness measurements"

[Reject] We prefer to keep "point thicknesses", since that is how they are referred to in the previous sentence. We would rather not create a false dichotomy by suggesting that spatial means are not also "measurements".

> [Page 17, Line 6-8] This sentence needs editing, "glacier and elevation band thicknesses" is not clear, probably refers to database categories?

[Accept] Changed "glacier and elevation band thicknesses" to "glacier and elevation band means", to clearly reference the categories set out earlier in the revised paragraph (see General comments) where the direct link is now made to each database table.

> [Page 17, Line 8-11] This sentence also needs editing, uncertainties are not "correct" but possible "realistic", "deemed to outweigh any benefit gained from averaging out random errors" also not clear and needs clarification

[Accept] Changed "correct" to "realistic" and reworded entire sentence (see revised paragraph in General comments).

> [Page 17, Line 10-11] this sentence is not clear, not clear how a distribution on Figure provides estimate of uncertainty. "thickness type" is also not a good term, refer to database categories

[Accept] Removed sentence (see revised paragraph in General comments).

> [Page 18, Figure 11] is not very useful and needs better explanation and editing of figure caption, suggest to delete figure and convey information of the figure in text and safe space

[Accept] Removed figure (see revised paragraph in General comments).

> [Page 18, Line 5] suggest to edit section title

[Reject] We think that "Data management > Error detection" is an appropriate title for this section. What would you suggest?

> [Page 19, Line 5] "number of changes" what is meant here, corrections in the database or additional input?

[Accept] Inserted "Every change, addition, and subtraction made to a file in the data package is tracked by git".

We mean any change made to any of the files in the data package, including corrections to existing data, the addition of new data, etc.

> [Page 19, Line 5] suggest to add "will" before "grow"

[Accept] Changed "git repositories inevitably grow in size" to "the git repository will inevitably grow in size".